# A protease-activatable luminescent biosensor and reporter cell line for authentic SARS-CoV-2 infection

Pehuén Pereyra Gerber[1,2], Lidia M. Duncan[1,2], Edward JD Greenwood[1,2], Sara Marelli[1,2], Adi Naamati[1,2], Ana Teixeira-Silva[1,2], Thomas WM Crozier[1,2], Ildar Gabaev[1,2], Jun R. Zhan[1,2], Thomas E. Mulroney[3], Emily C. Horner[3], Rainer Doffinger[4], Anne E. Willis[3], James ED Thaventhiran[1,3], Anna V. Protasio[5], Nicholas J. Matheson[1,2,6]*

1 Department of Medicine, University of Cambridge, Cambridge, United Kingdom, 2 Cambridge Institute for Therapeutic Immunology and Infectious Disease (CITIID), Jeffrey Cheah Biomedical Centre, University of Cambridge, Cambridge, United Kingdom, 3 MRC Toxicology Unit, University of Cambridge, Cambridge, United Kingdom, 4 Department of Clinical Biochemistry and Immunology, Cambridge University Hospitals NHS Foundation Trust, Cambridge, United Kingdom, 5 Department of Pathology, University of Cambridge, Cambridge, United Kingdom, 6 NHS Blood and Transplant, Cambridge, United Kingdom

* njm25@cam.ac.uk

**Data Availability Statement:** All relevant data are within the manuscript and its Supporting Information files. Availability of materials: Biological materials from this study are available from

## Abstract

Efforts to define serological correlates of protection against COVID-19 have been hampered by the lack of a simple, scalable, standardised assay for SARS-CoV-2 infection and anti-body neutralisation. Plaque assays remain the gold standard, but are impractical for high-throughput screening. In this study, we show that expression of viral proteases may be used to quantitate infected cells. Our assays exploit the cleavage of specific oligopeptide linkers, leading to the activation of cell-based optical biosensors. First, we characterise these biosensors using recombinant SARS-CoV-2 proteases. Next, we confirm their ability to detect viral protease expression during replication of authentic virus. Finally, we generate reporter cells stably expressing an optimised luciferase-based biosensor, enabling viral infection to be measured within 24 h in a 96- or 384-well plate format, including variants of concern. We have therefore developed a luminescent SARS-CoV-2 reporter cell line, and demonstrated its utility for the relative quantitation of infectious virus and titration of neutralising antibodies.

## Author summary

Techniques for measuring infection with SARS-CoV-2 in the laboratory are laborious and time-consuming, and different laboratories use different approaches. There is therefore no generally agreed way to quantitate neutralising antibodies against SARS-CoV-2, which block infection with the virus and protect people from COVID-19. In this study, we describe a new way to measure SARS-CoV-2 infection, which is much simpler and faster than existing methods. It relies on the production of a specific protease enzyme by the virus, which is able to cleave and activate an engineered protein biosensor in infected cells. This biosensor emits light in the presence of viral infection, and the amount of light released is used as a readout for the amount of infectious SARS-CoV-2 present. The signal

specified commercial sources, or from the corresponding author on execution of an appropriate Material Transfer Agreement (MTA). Clone B7 reporter cells will also be made available via the National Institute for Biological Standards and Control (NIBSC) repository (https://www.nibsc.org/).

**Funding:** This work was supported by the Medical Research Council (https://mrc.ukri.org/; Clinician Scientist Fellowship ref. MR/P008801/1 to NJM), NHS Blood and Transplant (https://www.nhsbt.nhs.uk/; grant ref. WPA15-02 to NJM), Addenbrooke's Charitable Trust (https://www.act4addenbrookes.org.uk/; grant ref. to 900239 NJM), the Rosetrees Trust (https://rosetreestrust.co.uk/; grant ref. G103718 to JRZ) and the National Institute for Health Research Cambridge Biomedical Research Centre (https://cambridgebrc.nihr.ac.uk/). The funders had no role in study design, data collection and analysis, decision to publish, or preparation of the manuscript.

**Competing interests:** The authors have declared that no competing interests exist.

is very sensitive, so the number of infected cells required is very small, and the method can be scaled-up to test many samples at once. In particular, we demonstrate how it can be used to detect different variants of SARS-CoV-2, and quantitate neutralising antibodies against these viruses.

## Introduction

Severe respiratory syndrome coronavirus 2 (SARS-CoV-2) infection or vaccination with spike protein or mRNA are able to induce neutralising antibodies and prevent coronavirus disease 2019 (COVID-19). Recently, SARS-CoV-2 variants have emerged with the ability to escape antibody neutralisation *in vitro*, and protective immunity *in vivo* [1]. Defining serological correlates of protection against these variants has been complicated by variability in the assays used to measure neutralising antibody titres [2]. Plaque Reduction Neutralisation Tests (PRNTs) remain the gold standard, but their utility for large-scale screening of sera is limited by low throughput and long turnaround times. Fluorescent [3] and luminescent [4] SARS-CoV-2 reporter viruses have been described, but the reverse genetic modification of SARS-CoV-2 is laborious, and these approaches are restricted to re-engineered reference strains. Accordingly, lentiviral particles pseudotyped with SARS-CoV-2 spike proteins are commonly used as surrogates, but these assays are not standardised between laboratories, and correlate imperfectly with results for wildtype virus [5]. There is therefore an urgent unmet need for a simple, high-throughput method to quantitate infection and neutralisation of authentic SARS-CoV-2 isolates, including new variants of concern.

Cell-based optical reporter systems have been instrumental in the response to a previous global pandemic, caused by the human immunodeficiency virus-1 (HIV-1). In particular, luciferase-based neutralisation assays using TZM-bl reporter cells are employed worldwide for the assessment of vaccine-elicited neutralising antibodies, monoclonal antibodies and the neutralising antibody response to HIV-1 infection [6,7]. Cited by >1000 publications, they have also been instrumental in the development of antiviral therapeutics, and facilitated a wealth of fundamental research on retrovirology. In this study, we therefore aim to develop an equivalent luminescent reporter cell line for SARS-CoV-2.

During the SARS-CoV-2 replication cycle, the 30 kb single-stranded positive-sense genomic RNA is used as a template to generate the polyproteins 1a and 1ab (pp1a and pp1ab). In turn, these polyproteins are processed into 16 non-structural proteins (nsp1 to nsp16) by the action of two virally-encoded proteases on sequence-specific cleavage sites: Papain-like Protease (PLPro, or nsp3), which cleaves nsp1, nsp2, and nsp3, and Main or 3C-like Protease (MPro, or nsp5), which cleaves the remaining non-structural proteins [8]. Both proteases contribute to the assembly of the viral replication and transcription complex (RTC), making them attractive targets for drug development. At the same time, we show here that the expression of SARS-CoV-2 protease activity during viral replication may be exploited for the detection and quantitation of infected cells, and demonstrate the utility of this approach for assays of candidate antivirals and neutralising antibodies.

## Results

### Detection of recombinant SARS-CoV-2 protease activity using FlipGFP-based reporters

The "flip" GFP (FlipGFP) reporter comprises an inactive, split eGFP molecule, in which cleavage of an oligopeptide linker is able to restore GFP fluorescence [9]. To test the feasibility of

protease-activatable biosensors for SARS-CoV-2, we therefore first replaced a tobacco etch virus (TEV) protease cleavage site present in the FlipGFP oligopeptide linker with candidate SARS-CoV-2 MPro (WT3c and Opt3c) or PLPro (PLP1-3) cleavage sites (**Fig 1A**), focussing on sequences conserved across SARS-CoV-2 viral isolates (**S1A Fig**). Candidate biosensors were co-transfected with blue fluorescent protein (BFP) plus/minus cognate SARS-CoV-2 protease into HEK293T cells, and analysed by flow cytometry after 24 h (**Fig 1B–1D**). To control for background FlipGFP fluorescence, mCherry was co-expressed with FlipGFP using a T2A peptide linker, and the ratio of FlipGFP/mCherry fluorescence in BFP+ (transfected) cells used to quantitate reporter activation (**S2A and S2B Fig**). All reporters showed an increase in FlipGFP fluorescence, but the magnitude of effect was variable (**Fig 1B**). We therefore selected the Opt3c-FlipGFP (**Fig 1C**, MPro) and PLP2-FlipGFP (**Fig 1D**, PLPro) biosensors for further evaluation. Similar results were obtained using epifluorescence microscopy (**Fig 1E**).

To confirm the specificity of these biosensors for their cognate protease, the same co-transfection assay was used to measure Opt3c-FlipGFP, PLP2-FlipGFP and TEV-FlipGFP activation in pair-wise combination with MPro, PLPro or TEV protease. In each case, an increase in FlipGFP fluorescence was observed only in the presence of cognate protease (**Fig 1F**). As a further check, we generated non-cleavable mutants of Opt3c-FlipGFP and PLP2-FlipGFP, by changing key residues in their respective cleavage sites. Neither non-cleavable control showed an increase in FlipGFP fluorescence in the presence of cognate protease (**Fig 1G**). Taken together, these results confirm that the selected SARS-CoV-2 biosensors are activated in a strictly protease and sequence-dependent manner.

## Activation of FlipGFP-based reporters by SARS-CoV-2 infection

To test whether Opt3c-FlipGFP and PLP2-FlipGFP could be activated by viral protease expression during SARS-CoV-2 infection, we made use of a permissive HEK293T cell line over-expressing ACE2 and furin [10]. These HEK293T-ACE2 cells are readily transfectable with reporter constructs, and form large syncytia within 24 h of SARS-CoV-2 infection (**S3A Fig**). Whilst syncytia are lost during flow cytometric analysis (**S3B and S3C Fig**), their formation has previously been exploited for the quantitation of SARS-CoV-2 infection by high-content imaging [11]. We therefore used fluorescent microscopy to analyse changes in FlipGFP fluorescence. HEK293T-ACE2 cells were transfected with reporter constructs, then infected after 12 h with SARS-CoV-2. After a further 24 h incubation, reporter activation was quantitated as the ratio of FlipGFP/mCherry fluorescence in spike-positive syncytia, compared with uninfected cells. As expected, a consistent increase in FlipGFP fluorescence was observed in spike-positive syncytia transfected with either Opt3c-FlipGFP or PLP2-FlipGFP, but not their corresponding non-cleavable controls (**Fig 2A and 2B**).

These fluorescent biosensors therefore provide proof-of-concept data confirming that viral protease activity may be used to signal SARS-CoV-2 infection. Nonetheless, the magnitude of effect is markedly reduced compared with over-expression of recombinant viral proteases (compare **Fig 2B** with **Fig 1G**). This likely reflects lower levels of protease expression during viral infection (**Fig 2C**). In addition, it is possible that the localisation of proteases and/or presence of other viral proteins and endogenous polyprotein substrates during authentic viral infection reduces their likelihood of encountering reporter molecules.

## Luciferase-based biosensors of authentic SARS-CoV-2 infection

To enzymatically amplify the signal from viral protease activity, we next sought to generate equivalent luciferase-based biosensors. The 30F-GloSensor reporter comprises an inactive, circularly permuted firefly luciferase (FFluc) molecule, in which cleavage of a specific

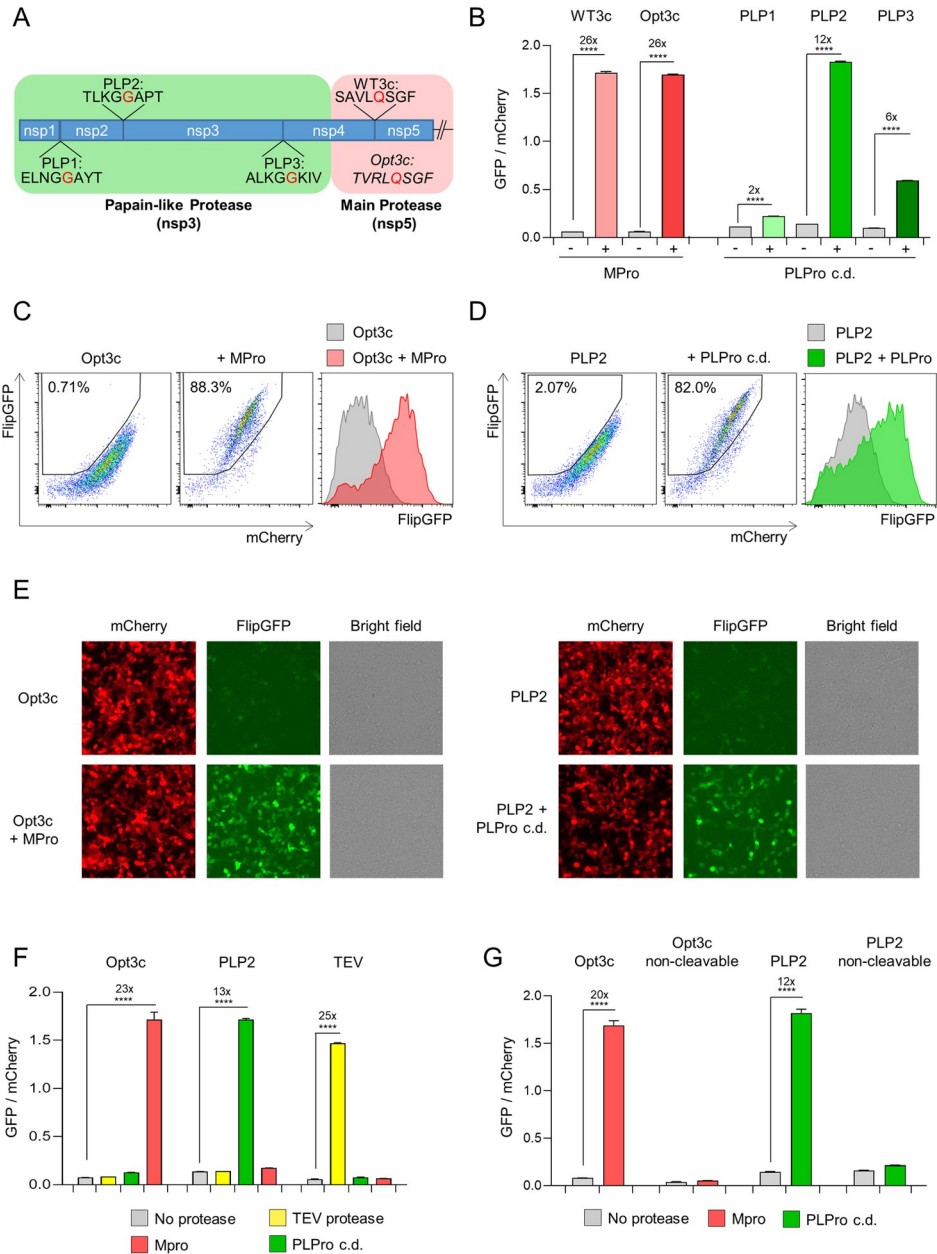

**Fig 1. Cell-based biosensors of SARS-CoV-2 protease activity.** (**A**) Diagram of pp1a polyprotein showing candidate SARS-CoV-2 cleavage sequences for Papain-like Protease (PLPro; PLP1-3) and Main Protease (MPro; WT3c and Opt3c). Further details on sequence selection are available in the **Materials and methods**. Sites of cleavage are highlighted in red (C-terminal side of indicated amino acid). (**B-D**) Activation of FlipGFP-based reporters by recombinant SARS-CoV-2 protease expression. HEK293T cells were co-transfected with BFP and indicated FlipGFP-based reporter constructs encoding candidate MPro or PLPro cleavage sequences ± MPro, PLPro c.d., or empty pcDNA3.1. Illustrative flow cytometry data for Opt3c-FlipGFP (**C**) and PLP2-FlipGFP (**D**) are shown. (**E**) Detection of FlipGFP-based reporter activation by epifluorescence microscopy. HEK293T cells were co-transfected with Opt3c-FlipGFP biosensor plus/minus MPro (left panel) or PLP2-FlipGFP biosensor plus/minus PLPro (right panel). FlipGFP and mCherry fluorescence were analysed by epifluorescence microscopy 24 h post-transfection. mCherry, red. FlipGFP, green. Representative of 3 independent experiments. (**F**) Protease-specificity of FlipGFP-based reporters. HEK293T cells were co-transfected with BFP and indicated FlipGFP biosensors ± MPro, PLPro c.d., TEV protease or empty pcDNA3.1. (**G**) Sequence specificity of FlipGFP-based reporters. HEK293T cells were co-transfected with BFP and indicated FlipGFP biosensors ± MPro, PLPro or empty pcDNA3.1. Further details on non-cleavable mutants are available in the **Materials and methods**. For all flow cytometry experiments (**B-D** and **F-G**), FlipGFP and mCherry fluorescence in BFP+ cells were analysed 24 h post-transfection. An indicative gating strategy is shown in **S2A Fig**,

with the % BFP+ cells in **S2B Fig**. Mean values ± SEM are shown for experiments performed in triplicate, representative of at least 3 independent experiments. **** p<0.0001. MPro, recombinant SARS-CoV-2 Main Protease. PLPro c.d., catalytic domain of recombinant SARS-CoV-2 Papain-Like Protease. TEV, recombinant TEV protease.

oligopeptide linker is able to restore luminescence [12]. Co-expression with Renilla luciferase (Rluc) from the same vector allows reporter levels and cell viability to be normalised between conditions. We therefore generated 30F-GloSensor constructs with the same sequence-specific oligopeptide linkers as our FlipGFP-based reporters, together with the equivalent non-cleavable controls. These constructs were co-transfected with/without cognate SARS-CoV-2 protease into HEK293T cells, and luminescence analysed after 24 h (**Fig 3A**). Normalising for Rluc luminescence, a 139-fold increase in luminescence was observed for the 30F-Opt3c biosensor (MPro), and a 74-fold increase for the 30F-PLP2 biosensor (PLPro)–much greater than the changes in fluorescence with FlipGFP-based reporters (compare **Fig 3A** with **Fig 1G**).

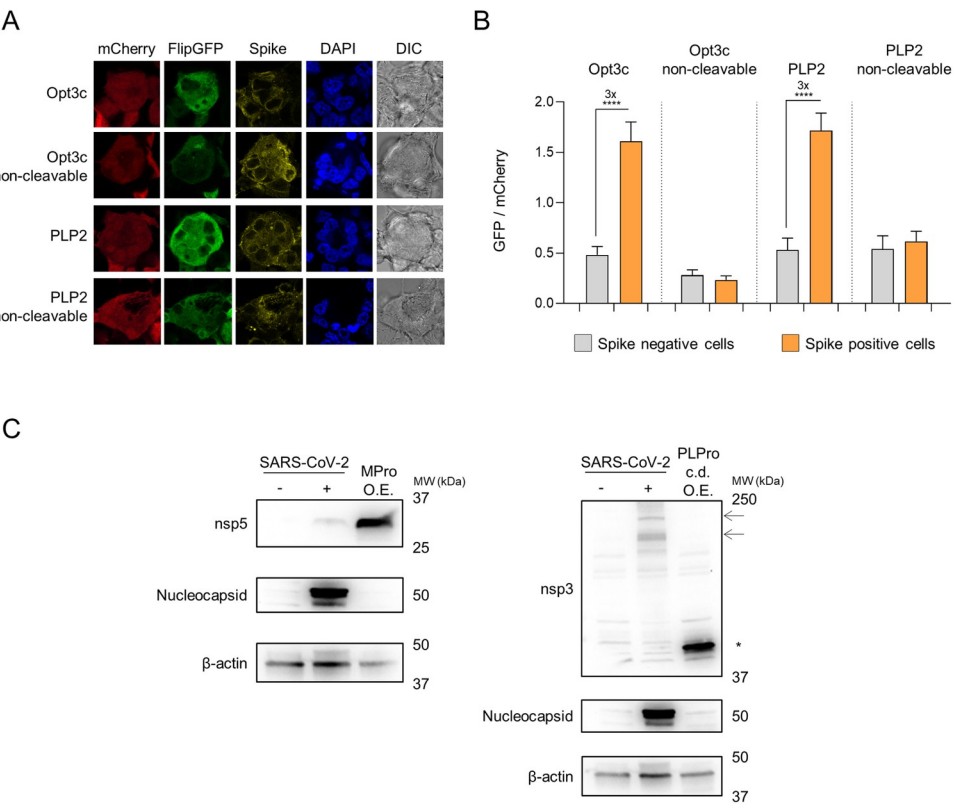

**Fig 2. Activation of FlipGFP-based reporters by SARS-CoV-2 infection.** (**A-B**) HEK293T-ACE2 cells were transfected with indicated FlipGFP biosensors, incubated for 12 h, then infected with SARS-CoV-2 at MOI = 1. Cells were fixed, permeabilised and stained for SARS-CoV-2 spike protein 24 h post-infection. FlipGFP and mCherry fluorescence in spike+ syncytia vs. spike- cells were analysed by confocal microscopy. Illustrative microscopy data from spike+ syncytia (**A**) and mean values ± SEM for at least 15 syncytia/cells from each condition (**B**) are shown, representative of 2 independent experiments. **** p<0.0001. mCherry, red. FlipGFP, green. Spike, yellow. DAPI, blue. DIC, differential interference contrast. (**C**) Expression levels of SARS-CoV-2 proteases. HEK293T-ACE2 cells were mock-infected, infected with SARS-CoV-2 at MOI = 1 or transfected with MPro (left panel) or PLPro (right panel) under standard conditions. Cells were lysed after 24 h and analysed by immunoblot using antibodies specific for nsp5 (MPro), the PLPro c.d. of nsp3 or nucleocapsid. β-actin was included as a loading control. Arrows indicate presumed full-length nsp3 (215 kDa) and intermediate cleavage products present in SARS-CoV-2-infected cells. Asterisk indicates recombinant PLPro c.d.. Representative of 2 independent experiments. O.E., over-expression. MPro, recombinant SARS-CoV-2 Main Protease. PLPro c.d., catalytic domain of recombinant SARS-CoV-2 Papain-Like Protease.

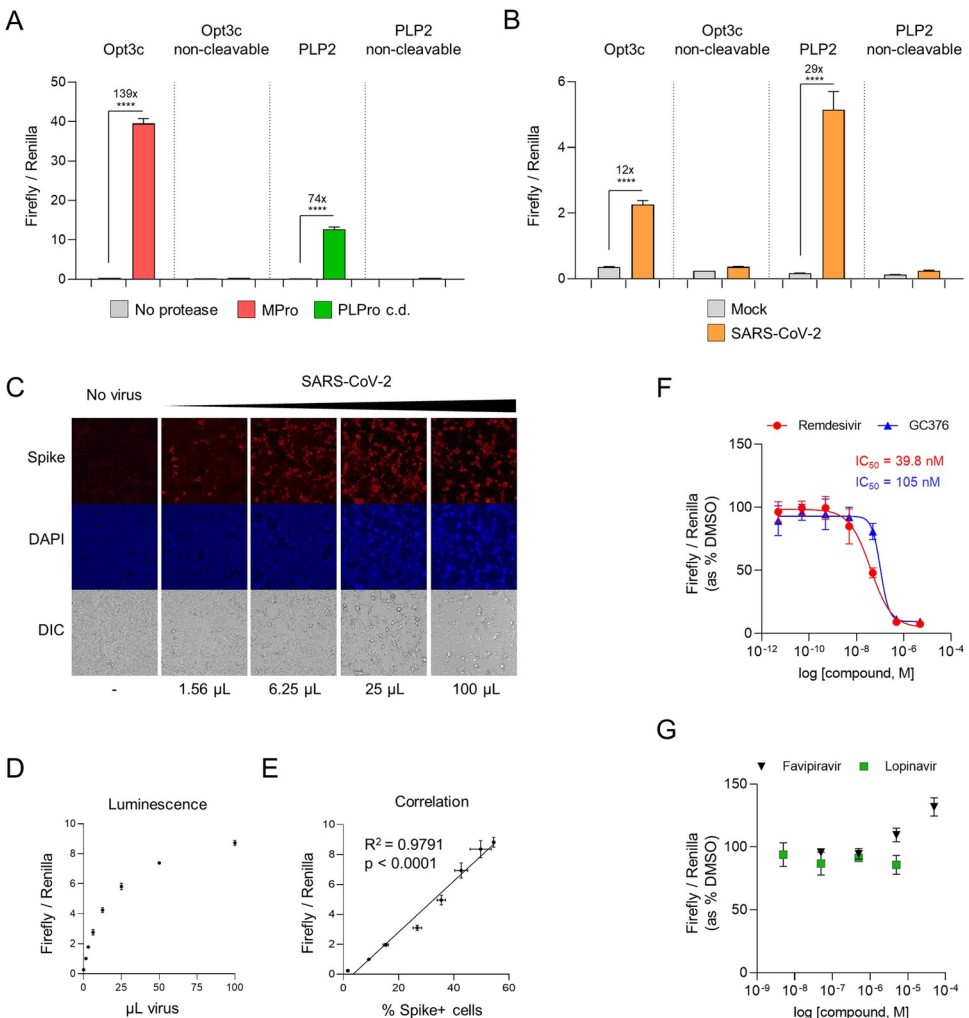

**Fig 3. Luminescent biosensors of authentic SARS-CoV-2 infection.** (**A**) Activation of luciferase-based reporters by recombinant SARS-CoV-2 protease expression. HEK293T cells were co-transfected with indicated 30F biosensors ± MPro, PLPro c.d., or empty pcDNA3.1. (**B**) Activation of luciferase-based reporters by SARS-CoV-2 infection. HEK293T-ACE2 cells were transfected with indicated 30F biosensors, incubated for 12 h, then mock-infected or infected with SARS-CoV-2 at MOI = 0.01. (**C-E**) Quantitation of infected cells. HEK293T-ACE2 cells were transfected with the 30F-PLP2 biosensor, incubated for 12 h, then mock-infected or infected with increasing doses of SARS-CoV-2. 1 µL of viral stock corresponds to MOI≈0.01. Cells were analysed in parallel 24 h post-infection by either epifluorescence microscopy for SARS-CoV-2 spike protein (**C**), or luminometry for Firefly and Renilla luciferase activities (**D**). Spike+ cells were enumerated by automated microscopy (Cellomics). Illustrative microscopy data (**C**) and mean values ± SEM (**D**) are shown for an experiment performed in duplicate (microscopy) or triplicate (luminometry). The correlation between the ratio of Firefly/Renilla luminescence and the proportions of spike+ cells is shown in **E**. Spike, red. DAPI, blue. DIC, differential interference contrast. $R^2$, Pearson's correlation coefficient. Representative of 2 independent experiments. (**F-G**) Inhibition of SARS-CoV-2 replication by candidate antivirals. HEK293T-ACE2 cells were transfected with the 30F-PLP2 biosensor, incubated for 12 h, then infected with SARS-CoV-2 at MOI = 0.01 in the presence of DMSO or decreasing doses of candidate antivirals. Titration curves and IC50s are shown for remdesivir and GC376. Representative of 2 independent experiments. For all experiments, Firefly and Renilla luciferase activities were measured by luminometry 24 h post-transfection (**A**) or infection (**B-G**). Unless otherwise stated, mean values ± SEM are shown for experiments performed in triplicate, representative of at least 3 independent experiments. For **F-G**, Firefly/Renilla luminescence is shown as % luminescence in the DMSO condition. **** $p<0.0001$. MPro, recombinant SARS-CoV-2 Main Protease. PLPro c.d., catalytic domain of recombinant SARS-CoV-2 Papain-Like Protease.

To test whether 30F-Opt3c and 30F-PLP2 could also be efficiently activated by viral protease expression during SARS-CoV-2 infection, HEK293T-ACE2 cells were again transfected with reporter constructs, then infected after 12 h with SARS-CoV-2. After a further 24 h incubation, reporter activation was quantitated as the ratio of FFluc/Rluc luminescence. Compared with uninfected cells, a marked increase in luminescence was observed in infected cells transfected with either 30F-Opt3c or 30F-PLP2, but not non-cleavable controls (**Fig 3B**). In the context of viral infection, 30F-PLP2 showed a better dynamic range (29-fold increase in luminescence) than 30F-Opt3c (12-fold increase in luminescence). We therefore selected 30F-PLP2 for further validation in assays of authentic SARS-CoV-2 virus.

First, we titrated a SARS-CoV-2 viral stock in HEK293T-ACE2 cells transfected with 30F-PLP2, and compared the luciferase signal with automated microscopy for spike. As expected, the ratio of FFluc/Rluc luminescence increased in accordance with the amount of virus, and correlated closely with the proportion of spike-positive (infected) cells (**Fig 3C–3E**). Next, HEK293T-ACE2 cells transfected with 30F-PLP2 were infected with SARS-CoV-2 in the presence of remdesivir, GC376, favipiravir, lopinavir or DMSO (**Fig 3F and 3G**). Similar to published data [13–15], these assays readily distinguished antiviral compounds abrogating SARS-COV-2 replication *in vitro* (remdesivir and GC376) from those that did not (favipiravir and lopinavir). IC50s for remdesivir and GC376 were in the low nM range, corresponding with previous results from PRNTs [13,14,16]. Importantly, since the luminescent signal was inhibited by remdesivir and GC376 –neither of which targets PLPro directly–biosensor activation must genuinely reflect viral replication.

## Generation of a luminescent SARS-CoV-2 reporter cell line for the relative quantitation of infectious virus and titration of neutralising antibodies

To simplify our assay and adapt it for large-scale serological screening, we optimised the 30F-PLP2/Rluc reporter cassette for lentiviral transduction (**S4 Fig**), and used it to generate a stable cell line expressing the 30F-PLP2 biosensor (**Fig 4A**). When these HEK293T-ACE2-30F-PLP2 cells were infected with wild-type SARS-CoV-2, a 20-fold increase in luminescence was seen at 24 h (**Fig 4B**). To further increase the dynamic range of the assay, we screened single-cell clones obtained from the bulk population of reporter cells. Clones B7 and G7 showed the maximum response to SARS-CoV-2 infection (>100-fold increase in FFluc/Rluc ratio at 24 h) (**Fig 4C**). Clone B7 was selected for the remaining experiments shown in this paper because it was morphologically identical to parental HEK293T cells, whereas clone G7 tended to grow in clumps. It will be made available to the community via the National Institute for Biological Standards and Control (NIBSC) repository (https://www.nibsc.org/).

Using clone B7 reporter cells, an increase in luminescence is readily detectable by 12 h post-infection, and the FFluc/Rluc ratio correlates closely with the frequency of spike-positive (infected) cells over a 24 h time course (**S5A–S5D Fig**). Because the luminescent signal at 24 h is a continuous readout dependent on both the starting inoculum and the rate of spreading infection (leading to an increase in the number of infected cells), the FFluc/Rluc ratio cannot be used directly for the absolute quantitation of infectious units in viral stocks (unlike a plaque assay). Nonetheless, assays may be conducted in a 96-well or 384-well plate format (**Fig 4D**), and–as expected based on PLP2 sequence conservation (**S1A Fig**)–cells are readily activated by different isolates of wildtype SARS-CoV-2, including the B.1.1.7 (alpha) and B.1.617.2 (delta) variants of concern (**Fig 4E**).

To demonstrate the utility of our luminescent reporter cell line for measuring SARS-CoV-2 neutralising activity, we tested serum samples from 5 healthy control donors 21 days after their first or second doses of Pfizer-BioNTech BNT162b2 mRNA vaccine using both clone B7

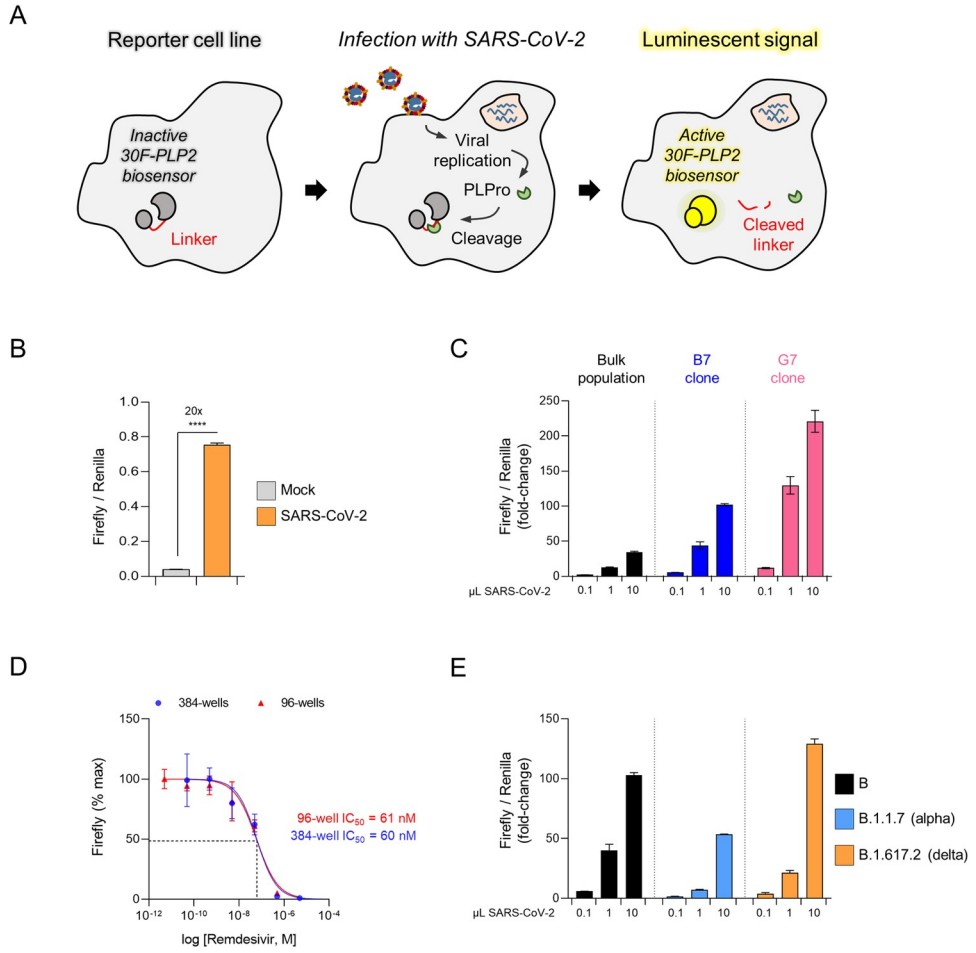

**Fig 4. An optimised luminescent SARS-CoV-2 reporter cell line.** (**A**) Schematic showing activation of luminescent reporter cell line by authentic SARS-CoV-2 infection. (**B**) Activation of reporter cells by SARS-CoV-2 infection. HEK293T-ACE2-30F-PLP2 cells (bulk population) were mock-infected or infected with SARS-CoV-2 at MOI = 0.01. (**C**) Selection of optimised cell lines. HEK2932-ACE2-30F-PLP2 cells (bulk population) or single-cell clones (B7 and G7) were mock-infected or infected with increasing doses of SARS-CoV-2. (**D**) Miniaturisation to 384-well plate format. Clone B7 reporter cells were infected with SARS-CoV-2 at MOI = 0.01 in the presence of decreasing doses of remdesivir in either a 96-well or 384-well plate format. Titration curves and IC50s for remdesivir are shown. Representative of 2 independent experiments. (**E**) Detection of SARS-CoV-2 variants of concern. Clone B7 reporter cells were mock-infected or infected with increasing doses of SARS-CoV-2 virus from the indicated lineages. Representative of 2 independent experiments. For **C** and **E**, 1 μL of viral stock corresponds to MOI≈0.01 for the lineage B and B.1.617.2 (delta) viral isolates, and MOI≈0.005 for the lineage B.1.1.7 (alpha) viral isolate. For all experiments, Firefly and Renilla luciferase activities were measured by luminometry 24 h post-infection. Unless otherwise stated, mean values ± SEM are shown for experiments performed in triplicate, representative of at least 3 independent experiments. For **C** and **E**, Firefly/Renilla luminescence is shown as fold-change with/without infection. For **D**, Firefly luminescence is shown as % maximum. **** $p < 0.0001$.

reporter cells and Plaque Reduction Neutralisation Tests (PNRTs) in VeroE6 cells (**Figs 5A–5C** and **S6**). Similar neutralisation curves were obtained from both assays (**Fig 5A**), with a striking correlation between the calculated neutralising titres at 50% inhibition (NT50s) (**Fig 5B**). Activation of reporter cells was stable over multiple passages (**S7A Fig**), and NT50s for control serum determined using these cells were highly reproducible across independent experiments and over time (**S7B Fig**).

Consistent with previous results [17], whilst anti-spike antibodies were readily detectable by immunoassay in all individuals (**S8 Fig**, middle and right panels), clear sigmoidal

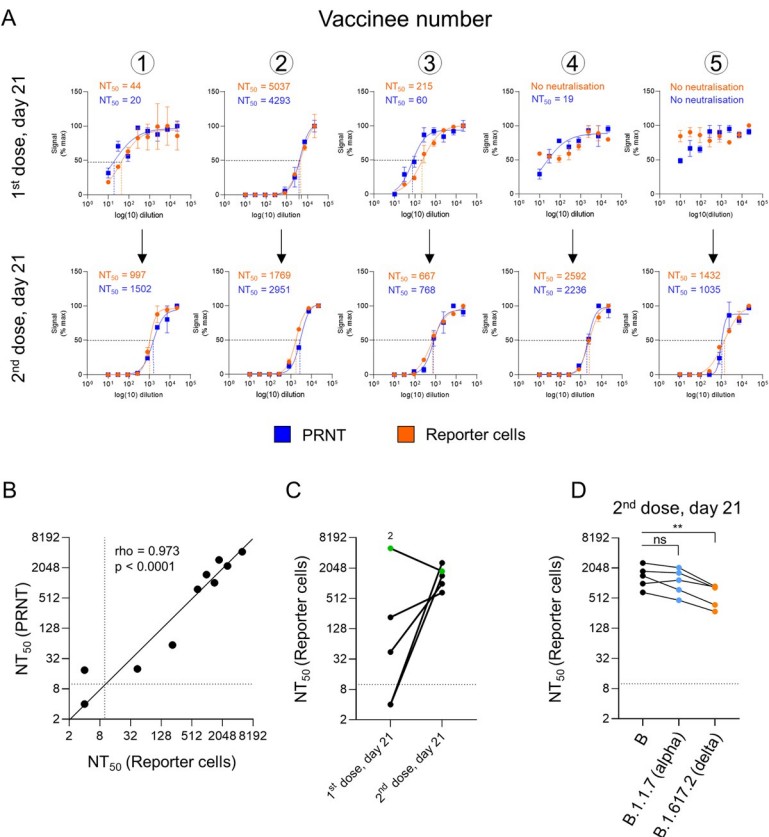

**Fig 5. Quantitation of neutralising activity in human serum samples. (A-D)** Measurement of neutralising antibody titres after inoculation with Pfizer-BioNTech BNT162b2 mRNA vaccine. (**A**) Neutralisation curves and NT50s for serum samples from 5 healthy control donors 21 days after their 1st (upper panels) or 2nd (lower panels) doses of BNT162b2 vaccine measured using either a Plaque Reduction Neutralization Test (PRNT, blue) or clone B7 reporter cells (orange). For PRNTs, SARS-CoV-2-ZsGreen viral stock (MOI = 0.001) was pre-incubated with a 3-fold dilution series of heat-inactivated serum before addition to VeroE6 cells. A semi-solid overlay was added after 2 h, and plaques enumerated by automated microscopy (Cellomics) 48 h post-infection. For the reporter cells, SARS-CoV-2 viral stock (MOI = 0.01) was pre-incubated with a 3-fold dilution series of heat-inactivated serum before addition to cells. Firefly and Renilla luciferase activities were measured by luminometry 24 h post-infection. Plaque number and Firefly/Renilla luminescence are shown as % maximum. Mean values ± SEM are shown for experiments performed in duplicate (PRNT) or triplicate (reporter cells), representative of 2 independent experiments. Illustrative data from PRNTs are included in **S6 Fig**. (**B**) Correlation between NT50s obtained for each sample (**A**) using either PRNTs (y-axis) or reporter cells (x-axis), with a solid black identity line (line of equality). Dotted lines indicate the limit of detection (lowest dilution) for each assay. rho, Spearman's rank correlation coefficient. (**C**) Comparison between NT50s for each donor after 1st and 2nd doses of BNT162b2 vaccine obtained using reporter cells (**A**). Samples with no detectable neutralising activity at the lowest dilution (dotted line) are plotted at an arbitrary NT50 of 4. Donor 2 is highlighted in green. (**D**) Measurement of neutralising antibody titres against lineage B.1.1.7 (alpha) and B.1.617.2 (delta) variants of concern. The same serum samples from healthy control donors obtained 21 days after their 2nd dose of BNT162b vaccine were processed as described in (**A**) but using lineage B.1.1.7 or B.1.617.2 viral stocks (MOI = 0.01). Representative of 2 independent experiments. Data for the lineage B.29 viral stock are as shown in **A-C**. ** $p < 0.01$.

neutralisation curves were observed in only 2/5 donors after their first dose (**Fig 5A**, upper panels). The highest NT50 was observed for donor 2, who also had detectable anti-nucleocapsid antibodies, suggesting previous SARS-CoV-2 infection (**S8 Fig**, left panel). Conversely, after the second dose, strong neutralising activity was seen for all donors tested (**Fig 5A**, lower panels, and **Fig 5B**). Finally, and again in agreement with previous observations [18], we observed a modest yet statistically significant reduction in neutralising activity against the B.1.617.2 (delta) variant of concern (**Fig 5D**).

## Discussion

Taken together, our data establish the feasibility of protease-activatable biosensors for the quantitation of authentic SARS-CoV-2 infection, and demonstrate the practical utility of a luciferase-based reporter cell line for the measurement of neutralising antibody activity in clinical samples. As well as serological surveys, they are ideally suited for both general virological assays, and screens of candidate antiviral compounds. Luminescent assays may be readily adapted to high-throughput platforms and, whilst the mechanism of reporter activation is different, the range of potential applications is similar to TZM-bl reporter cells for HIV infection.

Prior to the COVID-19 pandemic, protease-activatable reporters had been used successfully for the detection of recombinant proteases from a range of viruses including HIV-1, HCV, DENV-1, and MERS-CoV [19–22], but not endogenous viral proteases expressed during authentic infection. Whilst other groups have also described protease-activatable reporters for SARS-CoV-2 MPro [23–26], the primary aim of this study was to develop a simple, optical biosensor for authentic SARS-CoV-2 viral infection. Consistent with this, our luminescent reporter cell line allows assays to be conducted in 24 h, in a 96-well or 384-well plate format, without the need for sophisticated high-content microscopy.

Compared with FlipGFP biosensors, the window for luciferase-based biosensor activation was greatly increased. In addition, whilst the 30F-Opt3/MPro biosensor was readily activated by recombinant protease expression, the 30F-PLP2/PLPro biosensor performed markedly better during SARS-CoV-2 infection. This may reflect inversion of the relative expression levels of MPro and PLPro, or differential accessibility of the reporters (substrates) to viral proteases in the context of infection. Since PLPro of SARS-CoV has deubiquitylation (DUB) activity, and is able to regulate whole cell levels of ubiquitylation during viral replication [27], we suspect that PLPro of SARS-CoV-2 may also be able to readily access a wide variety of cellular proteins, including cytosolic reporters.

Unlike reverse-engineered fluorescent or luminescent reporter viruses, our reporter cells are able to detect different clinical SARS-CoV-2 isolates without modification of the viral genome, including emerging variants of concern. Beyond SARS-CoV-2, the PLP2 cleavage sequence is highly conserved in SARS-CoV, and partially conserved in MERS-CoV (**S1A Fig**). This suggests that the Papain-like proteases of these viruses will also be able to activate 30F-PLP2, and we confirmed this in the case of recombinant SARS-CoV PLPro (**S1B Fig**). Our protease-activatable luminescent biosensor and reporter cell line therefore provide an "off the shelf" solution for the quantitation of authentic betacoronavirus infection and neutralising antibody activity during this and (potentially) future coronavirus pandemics.

## Materials and methods

### Ethics statement

Ethical approval for this study was granted by the East of England–Cambridge Central Research Ethics Committee (08/H0308/176). Written informed consent was obtained from all volunteers prior to providing serum samples.

### Reagents

The following chemical compounds were used in this study: lopinavir (APExBIO, A8204); GC376 (a kind gift from Wayne Vuong, John C. Vederas and M. Joanne Lemieux); remdesivir (APExBIO, B8398); favipiravir (BioVision, 2778–5, a kind gift from Aartjan te Velthuis).

The following commercial assays and kits were used in this study: Dual-Glo Luciferase Assay System (Promega, E2920); NEBuilder HiFi DNA Assembly Cloning Kit (NEB, E5520S);

5-alpha Competent E. coli (NEB, C2987); Micro BCA Protein Assay Kit (Thermo Fisher, 23235).

The following antibodies were used in this study: anti-NSP3 (743–1072) SARS-CoV-2 (MRC PPU reagents, sheep DA126, 1st bleed, for immunoblot, 1:500); anti-NSP5 SARS-CoV-2 (MRC PPU reagents, sheep DA118, 2nd bleed, for immunoblot, 1:500); anti-SARS-CoV-2 spike [1A9] (Genetex, GTX632604, for microscopy and flow cytometry, 1:500); anti-SARS--CoV-2 nucleocapsid (Novus Biologicals, NB100-56683, for immunoblot, 1:1,000); anti-β-actin-HRP (Sigma, A5316, for immunoblot, 1:20,000); anti-mouse Alexa Fluor 647 (AF647) secondary (Jackson ImmunoResearch, #715-605-150, for microscopy and flow cytometry, 1:1,000); anti-mouse Alexa Fluor 594 (AF594) secondary (Jackson ImmunoResearch, #715-585-150, for microscopy, 1:1,000).

## Cell culture

HEK293T cells (a kind gift from Paul Lehner, authenticated by STR profiling [28,29]) and VeroE6 cells (a kind gift from Rupert Beale, authenticated by species-specific PCR (IDEXX BioAnalytics)) were cultured in DMEM supplemented with 10% fetal calf serum (FCS), 100 units/ml penicillin, and 0.1 mg/ml streptomycin at 37˚C in 5% CO2. All cells were regularly screened and confirmed to be mycoplasma negative (Lonza MycoAlert and IDEXX BioAnalytics).

## Vectors for transgene expression

**FlipGFP-based reporters.** To generate FlipGFP-based reporters, the TEV FlipGFP plasmid PCDNA3-FlipGFP(TEV cleavage seq) T2A mCherry (Addgene, #124429, a gift from Xiaokun Shu [9]) was used as a template for pairs of PCR reactions including primers designed to generate overlapping products replacing the TEV cleavage site with the indicated cleavage sequence (S1 Table). For example, to replace the TEV cleavage site with the PLP2 cleavage sequence, products generated by PCR reactions including 'Near AfeI'/'PLP2 Rv' and 'PLP2 Fw'/'Near AflII' primer pairs were used. The same plasmid was then digested with AfeI and AflII and assembled with the gel purified PCR products using the HiFi Assembly Master Mix (NEB, E5520).

For MPro, we selected a wildtype self-cleavage sequence (SAVLQ/SGF, herein termed WT3c) present between nsp4 and nsp5, and an "optimal" cleavage sequence (TVRLQ/SGF, herein termed Opt3c) found by substrate profiling of SARS-CoV MPro [30]. For PLPro, we selected all three cognate cleavage sequences present in the pp1a polyprotein (ELNGG/AYT, herein termed PLP1; TLKGG/APT, herein termed PLP2; and ALKGG/KIV, herein termed PLP3).

To generate a non-cleavable Opt3c-FlipGFP (MPro) reporter, the critical glutamine residue in the cleavage site was changed to isoleucine (TVRLI/SGF). For the PLP2-FlipGFP (PLPro) reporter, the critical LKGG sequence was scrambled to GLGK (TGLGK/APT).

**Luciferase-based reporters.** To generate circularly permuted firefly luciferase (FFluc)-based reporters, the pGloSensor-30F plasmid (Promega; discontinued, but available on request) was digested with BamHI and HindIII, then assembled (as above) with oligonucleotides encoding the indicated cleavage sequences (S2 Table).

To generate a lentiviral expression vector for the 30F-PLP2 reporter (S4 Fig), the pCMV-intron-30F-PLP2-pA-pSV40-hRluc reporter cassette was first amplified from the pGloSensor-30F PLP2, then assembled (as above) with pHRSIN-pCMV-EGFP-PGK-Puro [31] digested with EcoRI and NotI. Next, the SV40 poly(A) signal and promoter were replaced with an internal ribosome entry site (IRES) to generate the final pHRSIN-30F-PLP2-IRES-hRluc-

WPRE-PGK-Puro lentiviral expression vector. In brief, the pA-pSV40-hRluc fragment was excised by digestion with SalI, and replaced by three-way HiFi assembly (as above) with Renilla luciferase amplified by PCR from pGloSensor-30F and a 586 bp encephalomyocarditis virus (EMCV) IRES. The final sequence of the 30F-PLP2-IRES-hRluc-WPRE-PGK-Puro reporter cassette is shown (**S1 Appendix**).

**Viral proteases.** To generate an expression vector for SARS-CoV-2 Main protease (MPro), the open reading frame was amplified by PCR from the pDONR223 SARS-CoV-2 NSP5 plasmid (Addgene, #141259, a gift from Fritz Roth [32]), including an ATG start codon in the forward primer and a TAA stop codon in the reverse primer (MPro_Fw: GCTTGGTA CCGAGCTCGCACATGTCCGGATTCCGCAAGATG; MPro_Rv: GTGATGGATATCTG CAGTTACTGGAAGGTCACACCAGAGC), then assembled (as above) with EcoRI-linearised pcDNA3.1.

To generate an expression vector for the SARS-CoV-2 Papain-like protease catalytic domain (PLPro c.d.), we used the SARS-CoV nsp3 catalytic domain as a guide [33]. The coding sequence from amino acid 747E to 1061K was amplified by PCR from the pDONR207 SARS-CoV-2 NSP3 plasmid (Addgene, #141257, a gift from Fritz Roth[32]), including a Kozak sequence and ATG start codon in the forward primer and a TAA stop codon in the reverse primer (PLPcd_Fw: GCTTGGTACCGAGCTCGGCCACCATGGAGGTGAGGAC CATCAAGGTGTT; PLPcd_Rv: GTGTGATGGATATCTGCAGTTACTTGATGGTGGTG GTGTAGCT), then assembled (as above) with EcoRI-linearised pcDNA3.1. Where indicated, empty pcDNA3.1 was used as a control.

For expression of TEV protease, pcDNA3.1 TEV (full-length) (Addgene, #64276, a gift from Xiaokun Shu [34]) was used.

To generate an expression vector for the SARS-CoV Papain-like protease catalytic domain (PLPro c.d.) [33], a gene block was synthesized by GenScript (**S2 Appendix**), then assembled (as above) with pcDNA3.1 digested with BamHI and EcoRI.

**Other transgenes.** For expression of BFP, pTAG-BFP-N (Evrogen, FP172) was used.

To generate a lentiviral expression vector for human ACE2, the open reading frame was amplified from a HepG2 cDNA library by PCR using the indicated primers (ACE2_Fw: CGCCCGGGGGGGGATCCACTAGGTACCATGTCAAGCTCTTCCTGGCTCC; ACE2_Rv: CTAGAGTCGCGGCCGCTCTACTCGAGCTAAAAGGAGGTCTGAACATCATCAGT GTTTTG), then assembled (as above) with pHRSIN-pSFFV-GFP-PGK-Hygro [35] digested with KpnI and XhoI to generate pHRSIN-ACE2-Hygro.

For expression of human furin, pHRSIN-Furin-Puro (a kind gift from Paul Lehner) was used.

For lentiviral packaging, psPAX2 (Addgene, #12260, a gift from Didier Trono) and pCMV-VSV-G (Addgene, #8454, a gift from Bob Weinberg [36]) were used.

All constructs generated for this study were verified by Sanger sequencing (Source BioScience).

## Generation of HEK293T-ACE2 and HEK293T-ACE2-30F-PLP2 cells

For transduction of HEK293T cells with ACE2, a pHRSIN-ACE2-Hygro lentiviral stock was generated by co-transfection of HEK293T cells with psPAX2 and pCMV-VSV-G using standard methods. After selection with hygromycin for 2 weeks, cells were sorted for high cell surface ACE2 expression and single-cell cloned. Following expansion, a clone with stable, homogeneously high expression of ACE2 (herein termed clone 22) was selected for further experiments. For transduction with furin, a pHRSIN-Furin-Puro lentiviral stock was generated as above, and clone 22 cells (herein termed HEK293T-ACE2 cells) were selected with puromycin for a further 2 weeks. Finally, to generate HEK293T-ACE2-30F-PLP2 reporter cells, HEK293T-ACE2 cells were further transduced with a pHRSIN-30F-PLP2-IRES-hRluc-

WPRE-PGK-Puro lentiviral stock at a multiplicity of infection (MOI) of 3, generated as above. 100 single-cell clones were obtained by limiting dilution, then screened for sensitivity to SARS-CoV-2 infection (maximum increase in ratio of FFluc/Rluc luminescence).

## Production and titration of SARS-CoV-2 viral stocks

Unless otherwise stated, the virus used in this study was the lineage B viral isolate SARS-CoV-2/human/Liverpool/REMRQ0001/2020, a kind gift from Ian Goodfellow (University of Cambridge), isolated early in the COVID-19 pandemic by Lance Turtle (University of Liverpool) and David Matthews and Andrew Davidson (University of Bristol) from a patient from the Diamond Princess cruise ship [37–39].

Where specifically indicated, the following viruses were used: the lineage B.1.1.7 viral isolate SARS CoV-2 England/ATACCC 174/2020 (alpha variant), a kind gift from Greg Towers [40]; a lineage B.1.617.2 viral isolate (delta variant), a kind gift from Ravi Gupta [41]; and, for PNRTs, a lineage B fluorescent SARS-CoV-2 reporter virus encoding ZsGreen (SARS-CoV-2-ZsGreen, generated from an infectious full-length cDNA clone of SARS-CoV-2-Wuhan-Hu-1, but with a ZsGreen reporter gene inserted between orf7a and orf7b), a kind gift from Sam Wilson [42].

Viral stocks were typically prepared by passaging once in VeroE6 cells. In brief, cells were infected at a low MOI with the original viral stock, and incubated for 72 h (by which time cytopathic effect was evident). Virus-containing culture supernatants were then clarified by centrifugation at 600 g for 5 min, and immediately frozen in aliquots at -80˚C. Stocks of SARS-CoV-2-ZsGreen reporter virus were generated by transfection of the pCCI-4K-SARS-CoV-2-ZsGreen plasmid (GenBank: MW289908, https://mrcppu-covid.bio), prior to passaging once in VeroE6 cells as described. For determination of MOIs, viral stocks were titrated in VeroE6 cells by 50% tissue culture infectious dose (TCID50) assays using standard methods.

## Flow cytometric analysis of FlipGFP-based reporters in the presence of recombinant proteases

HEK293T cells were seeded at least 12 h in advance and transfected at approximately 50% confluency. Plasmids and TransIT-293 transfection reagent (Mirus) were mixed at a ratio of 1 μg DNA: 3 μLTransit-T293 in Opti-MEM (Gibco). For an experiment conducted in triplicate at a 48-well scale, we typically used (per condition): 150 ng BFP expression vector, 300 ng FlipGFP-based reporter construct, and 450 ng of either empty pcDNA3.1 (control) or pcDNA3.1 encoding the indicated protease; 2.7 μL TransIT-293; and 150 μL Opti-MEM (for 50 μL of transfection/well).

For titration of SARS-CoV-2 proteases, the total amount of DNA transfected was kept constant by adding additional empty pcDNA3.1. Where indicated, DMSO or candidate inhibitors of recombinant MPro or PLPro were added immediately after transfection.

Unless otherwise indicated, HEK293T cells were dissociated with trypsin 24 h post-transfection, resuspended in PBS + 0.5% FCS and analysed immediately by flow cytometry using a BD LSRFortessa equipped with 405 nm, 488 nm, 561 nm and 640 nm lasers. The ratio of FlipGFP/mCherry mean fluorescence intensity (MFI) in BFP+ (transfected) cells was used to quantitate reporter activation (FlowJo 10.7). An indicative gating strategy is shown in S2A Fig, with the % BFP+ cells for each experiment shown in S2B Fig.

## Flow cytometric analysis of spike protein expression in SARS-CoV-2-infected cells

HEK293T-ACE2 cells were seeded at a density of 9 x 10$^4$ cells/48-well in 250 μL complete media. The following morning, cells were infected with SARS-CoV-2 at MOI = 1 and incubated for 24 h.

To measure the fraction of infected cells using flow cytometry, cells were first dissociated with trypsin and fixed for 15 min by incubation in 4% paraformaldehyde (PFA). Cells were then permeabilised with Perm/Wash buffer (BD), stained for SARS-CoV-2 spike protein using a mouse monoclonal antibody (GeneTex, GTX632604) for 30 min at room temperature, washed twice, stained with an anti-mouse Alexa Fluor 647 (AF647) secondary antibody (Jackson ImmunoResearch, #715-605-150) for 30 min at room temperature, washed twice, resuspended in PBS + 0.5% FCS and analysed by flow cytometry as above.

## Automated microscopic analysis of spike protein expression in SARS-CoV-2-infected cells

HEK293T-ACE2 cells were seeded at a density 9 x $10^4$ cells/well of an 8-well μ-Slide (Ibidi, 80826) in 250 μL complete media. The following morning, cells were infected with SARS-CoV-2 at the indicated MOI and incubated for 24 h.

To measure the fraction of infected cells using automated microscopy, cells were first fixed for 15 min by incubation in 4% PFA. Cells were then permeabilised with Perm/Wash buffer (BD), stained for SARS-CoV-2 spike protein using a mouse monoclonal antibody (GeneTex, GTX632604) for 30 min at room temperature, washed twice, stained with an anti-mouse Alexa Fluor 594 (AF594) secondary antibody (Jackson ImmunoResearch, #715-585-150) for 30 min at room temperature, washed extensively, mounted with 200 μL/well of Fluoroshield Mounting Media (Sigma, F6057), and analysed by automated microscopy.

In brief, images were acquired using a Cellomics ArrayScan XTI high-throughput imaging platform (Thermo Fisher) using a 386 nm excitation/emission filter to detect DAPI-stained nuclei and a 560 nm excitation/emission filter to detect AF594. Images were then analysed with built-in high content HCS Studio software (by Thermo Fisher) using the Target Activation application. For this, cellular objects were identified by applying overlays (masks) based on DAPI intensity. Necessary steps to exclude non-cellular artefacts (large or small objects) were activated based on average nuclei size. Additionally, background correction was performed on both channels. The generated nuclei masks were then applied to the AF594 channel and the threshold for AF594 staining was determined using stained mock-infected cells. Finally, cells were considered infected if their AF594 signal was above this threshold. 42 fields were scanned for each sample/condition to ensure the analysis of a sufficient number of cells.

## Confocal microscopic analysis of FlipGFP-based reporters and spike protein expression in SARS-CoV-2-infected cells

HEK293T-ACE2 cells were seeded at a density 9 x $10^4$ cells/well of an 8-well μ-Slide (Ibidi, 80826) in 250 μL complete media. After 1 h, cells were transfected in duplicate with the indicated FlipGFP-based reporter constructs as above. The following morning, cells were infected with SARS-CoV-2 at the indicated MOI and incubated for 24 h.

To measure reporter activation in infected cells using confocal microscopy, cells were first fixed for 15 min by incubation in 4% PFA. Cells were then permeabilised with Perm/Wash buffer (BD), stained for SARS-CoV-2 spike protein using a mouse monoclonal antibody (GeneTex, GTX632604) for 30 min at room temperature, washed twice, stained with an anti-mouse AF647 secondary antibody (Jackson ImmunoResearch, #715-605-150) for 30 min at room temperature, mounted with 200 μL/well of Fluoroshield Mounting Media (Sigma, F6057), and analysed by confocal microscopy using a Zeiss LSM 710 Inverted confocal microscope equipped with 405, 458, 543 and 633 nm lasers and a Plan Apochromat 63X/1.40 Oil DIC M27 objective. For each reporter construct, the ratio of FlipGFP/mCherry MFI was calculated manually using Fiji (ImageJ) [43], by creating a mask around syncytiated cells that were both spike+ (infected) and mCherry+ (transfected).

## Immunoblotting

Washed cell pellets were lysed in PBS + 1% Triton supplemented with Halt Protease and Phosphatase Inhibitor Cocktail (Thermo Scientific) for 30 min on wet ice. Post-nuclear supernatants were heated in Laemelli Loading Buffer for 5 min at 95˚C, separated by SDS-PAGE and transferred to Immobilon-P membrane (Millipore). Membranes were blocked in PBS/5% non-fat dried milk (Marvel)/0.2% Tween and probed with the indicated primary antibody overnight at 4˚C. Reactive bands were visualised using HRP-conjugated secondary antibodies and SuperSignal West Pico or Dura chemiluminescent substrates (Thermo Scientific). Typically, 10–20 µg total protein was loaded per lane.

## Luminescent analysis of luciferase-based reporters in the presence of recombinant proteases

HEK293T cells were transfected in triplicate at a 48-well scale essentially as for transfection of FlipGFP-based reporter constructs, typically using 150 ng of the indicated pGloSensor-30F reporter construct and 150 ng of empty pcDNA3.1 (control) or pcDNA3.1 encoding the indicated protease.

After 24 h incubation, media was aspirated from each well, and cells lysed with 50 µL/well Dual-Glo Luciferase Buffer (Promega) diluted 1:1 with PBS + 1% NP-40, for 10 min at room temperature. Lysates were then transferred to opaque half-area 96-well plates, and reporter activation quantitated as the ratio of firefly luciferase (FFluc)/Renilla luciferase (Rluc) activity measured using the Dual-Glo kit (Promega) according to the manufacturer's instructions. In brief, FFluc activity was first measured using a ClarioStar microplate reader. 25 µL Stop and Glo Buffer and Substrate (Promega) was then added to each well. After incubation for 10 min at room temperature, Rluc activity was measured using the same ClarioStar microplate reader and the ratio of FFluc/Rluc activity calculated for each condition.

## Luminescent analysis of luciferase-based reporters in SARS-CoV-2-infected cells

HEK293T-ACE2 cells were reverse-transfected with plasmids and TransIT-293 at a ratio of 1 µg DNA: 3 µLTransit-T293 in Opti-MEM (Gibco). For an experiment conducted in triplicate at a 48-well scale, we typically used (per condition): 900 ng pGloSensor-30F reporter construct; 2.7 µL TransIT-293; and 150 µL Opti-MEM (for 50 µL of transfection mix/well). $2.7 \times 10^5$ cells were dissociated with Accutase, combined with the transfection mix, and seeded at $9 \times 10^4$ cells/well.

The following morning, cells were infected with SARS-CoV-2 at MOI = 0.01 (or for the titration experiments, with the indicated volume of viral stock) and incubated for 24 h. Media was aspirated from each well, and cells lysed with 50 µL/well of Dual-Glo Luciferase Buffer (Promega) diluted 1:1 with PBS + 1% NP-40 for 10 min at room temperature. Lysates were then transferred to opaque half-area 96-well plates, and reporter activation quantitated as the ratio of FFluc/Rluc activity as above. Where indicated, candidate antivirals were added to the cells 1 h before infection with SARS-CoV-2.

## Luminescent analysis of SARS-CoV-2-infected HEK293T-ACE2-30F-PLP2 reporter cells

Unless otherwise stated, HEK293T-ACE2-30F-PLP2 reporter cells (either the bulk population, or indicated clone) were seeded at a density of $4 \times 10^4$ cells/96-well in 100 µL complete media. The following morning, cells were infected with SARS-CoV-2 at MOI = 0.01 and incubated

for 24 h. Media was aspirated from each well, and cells lysed with 25 μL/well of Dual-Glo Luciferase Buffer (Promega) diluted 1:1 with PBS + 1% NP-40 for 10 min at room temperature. Lysates were then transferred to opaque half-area 96-well plates, and reporter activation quantitated as the ratio of FFluc/Rluc activity as above. For evaluation of candidate antivirals, compounds were added to the cells 1 h before infection with SARS-CoV-2. To measure SARS-CoV-2 neutralising antibody titres, SARS-CoV-2 viral stock (MOI = 0.01) was pre-incubated with a 3-fold dilution series of heat-inactivated serum for 2 h at 37˚C, prior to addition to reporter cells. Where indicated, assays were performed in 384-well plates, with cell numbers and reagent volumes scaled according to surface area, and reporter activation quantitated directly as FFluc activity.

## Plaque Reduction Neutralisation Tests (PRNTs)

PRNTs were performed essentially as previously described [44], but utilising the SARS-CoV-2-ZsGreen reporter virus. In brief, VeroE6 cells were seeded in 48-well plates to obtain a confluent monolayer at the time of infection. SARS-CoV-2-ZsGreen viral stock (MOI = 0.001) was pre-incubated with a 3-fold dilution series of the heat-inactivated serum for 2 h at 37˚C, before addition to the cells. A semi-solid overlay, made of 0.2% agarose in complete DMEM, was added after 2 h. 48 h post-infection, plates were fixed for 30 min in 4% PFA. Fixative was replaced with PBS containing a 1:5,000 dilution of DAPI, and ZsGreen fluorescence analysed by automated microscopy (Cellomics). 36 images were acquired for each well of a 48-well plate at a magnification of 5x, allowing full area coverage (1,728 images per 48-well plate). The 36 images for each well were stitched together using a Fiji (ImageJ) [43] macro, and the resulting 48 composite images stacked together in a single file from which the background was subtracted. Then, montages (6 rows, 8 columns) were generated for each stack and inverted. Illustrative PRNT data is shown in **S6 Fig**.

## Serological assessment

Serum samples from healthy control donors were obtained 21 days after their 1st or 2nd dose of Pfizer-BioNTech BNT162b2 mRNA vaccine, heat-inactivated at 56˚C for 30 min, then frozen in aliquots at -80˚C.

Antibodies to SARS-CoV-2 nucleocapsid, trimeric spike and spike receptor binding domain were quantitated by Luminex bead-based immunoassay as previously described [45]. In brief, recombinant SARS-CoV-2 proteins were covalently coupled to carboxylated bead sets (Luminex; Netherlands) to form a multiplex assay. Coupled bead sets were incubated with serum samples at a dilution of 1/100 for 1 h at room temperature, washed three times with 10mM PBS/0.05% Tween-20, incubated for 30 min with a PE-labelled anti–human IgG-Fc antibody (Leinco/Biotrend), washed again, then resuspended in 100 μl PBS/Tween-20. They were then analysed on a Luminex analyser (Luminex / R&D Systems) using Exponent Software V31. Specific binding was reported as mean fluorescence intensity (MFI). Serum samples from patients with PCR-confirmed COVID-19 were used as positive controls, and stored serum samples collected prior to November 2019 were used as negative controls. Diagnostic cut-offs were determined by receiver operating characteristic (ROC) curve analysis.

## SARS-CoV-2, SARS-CoV, and MERS-CoV sequence logos

A SARS-CoV-2 genomic alignment file was retrieved from the GISAID database (https://www.epicov.org/). At the time of access (30/06/2020), this alignment (msa_0630) contained 50,387 sequences. Genome sequences and alignments for MERS-CoV and SARS-CoV were accessed via the NCBI Virus portal (https://www.ncbi.nlm.nih.gov/labs/virus/vssi/#/virus?

SeqType_s = Nucleotide). For the MERS-CoV sequences, we searched the database for Virus with taxonid = 1335626 and Host with taxonid = 'human'. We set the nucleotide completeness option to 'complete' and alignments were generated using the NCBI alignment function. Consensus sequences automatically generated and included in the alignment file where removed. For the SARS-CoV sequences, we searched the database for Virus with taxonid = 694009 and Host taxonid = 'human'. Because of the large number of sequences (~15,000), it was not possible to perform the alignment using the NCBI alignment function. Instead, FASTA sequences were downloaded and aligned using MAFFT [46] with default parameters.

Having manually inspected the genomic alignments to identify the regions of interest, we used the 'extractalign' function in the European Molecular Biology Open Software Suite (EMBOSS) [47]. In the case of the GISAID alignment, we removed entries that were incomplete or that presented nucleotide ambiguities (non-CTGA bases). The resulting sets (one per region of interest and four per alignment inspected) were conceptually translated using the 'transeq' function in EMBOSS. The resulting amino acid sequences were used as input for the WebLogo application (http://weblogo.berkeley.edu) [48,49].

### Statistical analysis

General data manipulation was conducted using Microsoft Excel, and statistical analysis using Prism 8.0 (GraphPad). Unless otherwise stated, sample means were compared by one-way ANOVA followed by Tukey's multiple comparison test. To compare results of luminometry and automated microscopy, linear regression and Pearson's correlation coefficient were used. To calculate half-maximal inhibitory concentrations (IC50s), FFluc/Rluc ratios were analysed using the log(inhibitor) vs. response—Variable slope (four parameters) function.

To calculate neutralising antibody titres at 50% inhibition (NT50s), FFluc/Rluc ratios or plaque numbers were analysed using the Sigmoidal, 4PL, X is log(concentration) function. To compare results from PRNTs and reporter cells, Spearman's rank correlation coefficient was used. Differences in NT50s for lineage B, B.1.1.7 and B.1.617.2 viruses were compared by non-parametric one-way ANOVA followed by Dunn's multiple comparison test.

### Supporting information

**S1 Fig. Cleavage sequences of highly pathogenic human betacoronaviruses.** (**A**) Conservation of cleavage sequences. Amino acid sequence logos for cleavage sequences between nsp1/nsp2 (PLP1), nsp2/nsp3 (PLP2), nsp3/nsp4 (PLP3), and nsp4/nsp5 (WT3c) of SARS-CoV-2, SARS-CoV and MERS-CoV viruses. Relative letter heights indicate conservation across the sequences analysed. (**B**) Luminescent biosensor activation by PLPro of SARS-CoV. HEK293T cells were co-transfected with the 30F-PLP2 biosensor and the catalytic domain of recombinant Papain-Like Protease (PLPro c.d.) of either SARS-CoV-2 or SARS-CoV. Firefly and Renilla luciferase activities were measured by luminometry 24 h post-transfection. Mean fold-changes in Firefly/Renilla luminescence ± SEM in the presence or absence of protease are shown for an experiment performed in triplicate. Representative of 2 independent experiments.
(JPG)

**S2 Fig. Indicative gating strategy and transfection efficiency for flow cytometry experiments using FlipGFP-based reporters.** (**A**) Indicative gating strategy. Cells were typically gated on FSC-H and SSC-H, and doublets excluded using FSC-W. FlipGFP and mCherry fluorescence were analysed in BFP+ (transfected) cells. Data illustrate change in fluorescence of Opt3c-FlipGFP biosensor in the presence (lower panels) or absence (upper panels) of Main

Protease (MPro). (**B**) Transfection efficiency. % BFP+ (transfected) cells for experiments in **Fig 1B**, **1F and 1G** are displayed. Mean values ± SEM are shown for experiments performed in triplicate, representative of at least 3 independent experiments.
(JPG)

**S3 Fig. Infection of HEK293T-ACE2 cells by SARS-CoV-2.** (**A**) Cytopathic effect in HEK293T-ACE2 cells. HEK293T or HEK293T-ACE2 cells were infected with SARS-CoV-2 at MOI = 1 then examined for cytopathic effect using brightfield microscopy after 24 h. Representative of >10 independent experiments. (**B-C**) Loss of spike+ HEK293T-ACE2 cells during flow cytometric analysis. HEK293T-ACE2 cells were infected with SARS-CoV-2 at MOI = 1. Cells were analysed in parallel 24 h post-infection for SARS-CoV-2 spike protein by either automated microscopy (Cellomics) or flow cytometry. In each case, the proportion of spike + cells was measured. Illustrative data (**B**) and mean values ± SEM (**C**) are shown for an experiment performed in triplicate. **** $p < 0.0001$. Spike, yellow. DAPI, blue. Representative of 2 independent experiments.
(JPG)

**S4 Fig. 30F-PLP2 constructs.** Diagrams of 30F-PLP2 luciferase-based reporter in pGloSensor-30F and pHRSIN-30F-PLP2-IRES-hRluc-WPRE-PGK-Puro expression vectors. 30-FF-358-544/30-FF-4-354, circularly permuted firefly luciferase-based reporter, orange (numbers indicate positions of amino acids in wildtype FFluc). hRluc, codon-optimised (humanised) Renilla luciferase, cyan. LTR, HIV-1 long terminal repeat, yellow. IRES, internal ribosome entry site, brown. WPRE, woodchuck hepatitis virus post-transcriptional regulatory element (WPRE), purple. PuroR, puromycin resistance, green.
(JPG)

**S5 Fig. Kinetics of reporter cell activation.** (**A-D**) Clone B7 reporter cells were mock-infected or infected with SARS-CoV-2 at MOI = 0.01 and analysed in parallel at the indicated time points post-infection by either epifluorescence microscopy for SARS-CoV-2 spike protein (**A**), or luminometry for Firefly and Renilla Luciferase activities (**B**). Spike+ cells were enumerated by automated microscopy (Cellomics). Firefly/Renilla luminescence is shown as fold-change with/without infection. Mean values ± SEM (**A-C**) are shown for an experiment performed in triplicate, together with illustrative microscopy data (**D**). The correlation between the ratio of Firefly/Renilla luminescence and the proportions of spike+ cells is shown in **C**. Spike, red. DAPI, blue. $R^2$, Pearson's correlation coefficient. Representative of 2 independent experiments.
(JPG)

**S6 Fig. Illustrative Plaque Reduction Neutralisation Test (PRNT) data. (A-B)** Illustrative PRNT data from **Fig 5A**. SARS-CoV-2-ZsGreen viral stock (MOI = 0.001) was pre-incubated with a 3-fold dilution series of heat-inactivated serum from 5 healthy control donors 21 days after their 1st dose (**A**) or 2nd dose (**B**) of Pfizer-BioNTech BNT162b2 mRNA vaccine before addition to VeroE6 cells. A semi-solid overlay was added after 2 h, and cells fixed and analysed by automated microscopy (Cellomics) 48 h post-infection. For each well, 36 independent images were acquired and stitched together using Fiji (ImageJ). ZsGreen fluorescence is shown in grey.
(JPG)

**S7 Fig. Stability of clone B7 reporter cells. (A)** Comparison of luminescent biosensor activation in low passage vs. high passage reporter cells. The first freeze of clone B7 reporter cells (Passage 2) and cells kept in culture for two months (Passage 20) were infected with

SARS-CoV-2 at MOI = 0.01. Firefly and Renilla luciferase activities were measured by luminometry 24 h post-infection. Mean Firefly/Renilla luminescence ± SEM are shown for an experiment performed in triplicate. Representative of 2 independent experiments. (**B**) Reproducibility of NT50s obtained using clone B7 reporter cells. The serum from a healthy control donor obtained 21 days after their 2$^{nd}$ dose of Pfizer-BioNTech BNT162b2 mRNA vaccine was frozen in single-use (20 µL) aliquots at -80˚C and routinely included as an internal control in all assays of SARS-CoV-2 neutralising activity. NT50s were determined using clone B7 reporter cells as shown in **Fig 5A**. Data are shown for 9 independent experiments conducted in duplicate over a period of three months. The dotted line indicates the limit of detection (lowest dilution) for the assay.
(JPG)

**S8 Fig. Quantitation of SARS-CoV-2 antibodies in human serum samples by immunoassay.** Antibodies to SARS-CoV-2 nucleocapsid, trimeric spike and spike receptor-binding domain were quantitated by Luminex bead-based immunoassay. Negative controls, stored serum samples collected prior to 2019. Vaccinated, serum samples from 5 healthy control donors 21 days after their 1$^{st}$ dose of Pfizer-BioNTech BNT162b2 mRNA vaccine (same serum samples as **Fig 5A**, upper panels). Donor 2 is highlighted in green. Dotted lines indicate diagnostic cut-offs determined by receiver operating characteristic (ROC) curve analysis. MFI, mean fluorescence intensity.
(JPG)

**S1 Table. PCR primers for generation of FlipGFP-based reporters.**
(PDF)

**S2 Table. Oligonucleotides for generation of luciferase-based reporters.**
(PDF)

**S1 Appendix. Complete sequence of 30F-PLP2-IRES-hRluc-WPRE-PGK-Puro reporter cassette.**
(PDF)

**S2 Appendix. Gene block for cloning of SARS-CoV Papain-like protease catalytic domain.**
(PDF)

**S1 File. Numerical values used to generate figures.**
(XLSX)

## Acknowledgments

The authors thank Mark Wills for overseeing the Containment Level 3 facility, Nika Romashova for assistance with automated microscopy, Paul Lehner, James Nathan, Wayne Vuong, John Vederas, Joanne Lemieux, Aartjan te Velthuis, Rupert Beale, David Ho, Ian Goodfellow, Greg Towers, Sam Wilson, Petra Mlcochova and Ravi Gupta for supplying key reagents, and members of the Matheson and Lehner laboratories for critical discussion.

## Author Contributions

**Conceptualization:** Pehuén Pereyra Gerber, Lidia M. Duncan, Sara Marelli, Adi Naamati, Ildar Gabaev, Nicholas J. Matheson.

**Data curation:** Pehuén Pereyra Gerber, James ED Thaventhiran, Anna V. Protasio.

**Formal analysis:** Pehuén Pereyra Gerber, Rainer Doffinger, Anna V. Protasio.

**Funding acquisition:** James ED Thaventhiran, Nicholas J. Matheson.

**Investigation:** Pehuén Pereyra Gerber, Lidia M. Duncan, Edward JD Greenwood, Sara Marelli, Adi Naamati, Thomas E. Mulroney, Emily C. Horner, Rainer Doffinger.

**Methodology:** Pehuén Pereyra Gerber, Lidia M. Duncan, Edward JD Greenwood, Sara Marelli, Adi Naamati, Rainer Doffinger, Nicholas J. Matheson.

**Project administration:** Anne E. Willis, James ED Thaventhiran, Nicholas J. Matheson.

**Resources:** Ana Teixeira-Silva, Thomas WM Crozier, Jun R. Zhan, Anne E. Willis, James ED Thaventhiran, Nicholas J. Matheson.

**Software:** Anna V. Protasio.

**Supervision:** James ED Thaventhiran, Nicholas J. Matheson.

**Visualization:** Pehuén Pereyra Gerber, Anna V. Protasio, Nicholas J. Matheson.

**Writing – original draft:** Pehuén Pereyra Gerber, Nicholas J. Matheson.

**Writing – review & editing:** Pehuén Pereyra Gerber, Lidia M. Duncan, Edward JD Greenwood, Sara Marelli, Adi Naamati, Ana Teixeira-Silva, Thomas WM Crozier, Ildar Gabaev, Jun R. Zhan, Thomas E. Mulroney, Emily C. Horner, Rainer Doffinger, Anne E. Willis, James ED Thaventhiran, Anna V. Protasio, Nicholas J. Matheson.

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
