## [Decision Letter · Decision Letter 0]

11 Aug 2021

Dear Dr Matheson,

Thank you very much for submitting your manuscript "A protease-activatable luminescent biosensor and reporter cell line for authentic SARS-CoV-2 infection" for consideration at PLOS Pathogens. As with all papers reviewed by the journal, your manuscript was reviewed by members of the editorial board and by several independent reviewers. In light of the reviews (below this email), we would like to invite the resubmission of a significantly-revised version that takes into account the reviewers' comments.

Both reviewers appreciated the advance presented by this reporter cell line, but also raised concerns about the interpretation of the data as well as a need to confirm the use of this reporter cell line with a live virus assay. Some data which show how the metric reported by this reporter cell line corresponds in some way to more familiar metrics used in traditional PRNT or FFU reduction assays would broaden the impact and utility of this system for the community.   

We cannot make any decision about publication until we have seen the revised manuscript and your response to the reviewers' comments. Your revised manuscript is also likely to be sent to reviewers for further evaluation.

Sincerely,

Mehul Suthar

Associate Editor

PLOS Pathogens

Benhur Lee

Section Editor

PLOS Pathogens

Kasturi Haldar

Editor-in-Chief

PLOS Pathogens

orcid.org/0000-0001-5065-158X

Michael Malim

Editor-in-Chief

PLOS Pathogens

orcid.org/0000-0002-7699-2064

Reviewer's Responses to Questions

**Part I - Summary**

Reviewer #1: In this manuscript, Gerber et al. seek to develop a luminescent reporter cell line for the detection of replication of SARS-CoV-2 in susceptible cells. In doing so, they hope to produce a more reproducible and streamlined approach for the identification of therapeutics and the testing of serological correlates of protection. The authors reference the development of similar reporter systems that have been used in different viral models, including HIV, that have been broadly adopted.

Gerber et al. provide solid rationale for their experimental models including the evaluation of small molecule inhibitors Remdesivir and the evaluation of human polyclonal sera. Overall the manuscript provides a sufficient description of the process of developing the reporter cell line. Adequate controls are used to support the efficacy, specificity, and sensitivity of the reporters, and statistical models used seem appropriate.

Reviewer #2: The authors present proof-of-principle experiments using the two SARS-CoV-2-encoded proteases (PLPro/nsp3 and MPro/nsp5) to activate fluorescent and luminescent reporter constructs.

Their fluorescent reporter constructs based on FlipGFP showed a specific increase in GFP fluorescence 24h after co-transfection with viral protease, which was quantifiable by flow cytometry as well as microscopy. However, the use of this biosensor to detect authentic SARS-CoV-2 infection was limited by the tendency of infected cells to form large syncytia thus eliminating flow cytometry as a possible read-out, as well as a reduction in effect (3x during viral infection compared to 26/12x for proof-of-principle) most likely due to lower levels of protease expression during viral infection.

The authors therefore focus on a luminescent construct with enzymatic amplification of signal to circumvent the lower levels of protease activity during authentic SARS-CoV-2 infection. This system shows stronger induction during natural infection, with the PLP2 construct showing greater signal increase compared to Opt3c (29x compared to 12x).

The authors further validate the 30F-PLP2 construct by titration as well as in the presence of antiviral compounds and proceed to generate a stable reporter cell line via lentiviral transduction followed by clonal selection. This yields a versatile reporter cell line that allows detection of viral infection (both SARS-CoV-2 wildtype and B1.1.7 and thus likely other VOCs) within 24 hours in both 96- and 384-well format that can be used for measuring neutralising antibodies.

In summary, the protease-activatable luminescent reporter cell line presents a valuable tool for the evaluation of neutralising antibody activity that can be performed in a quantitative high-throughput setting and is likely applicable to a wider range of current and future SARS-CoV-2 variants.

**Part II – Major Issues: Key Experiments Required for Acceptance**

Reviewer #1: However, while overall this manuscript is strong, there are some concerns that should be addressed.

In the development of the initial strategy, what percentage of cells are transfected in fig 1 (BFP+)? The data and gating strategy should be shown as supplemental data. In later studies the luciferase system is moved to a lentivirus approach and stable cells are generated. How stable is the luciferase activity over time and passage number? These impact feasibility and reproducibility a key goal of the manuscript.

Additionally, in panel E of figure 4, the variant of concern B.1.1.7 is detected using the cells. In order to support the statement that the luminescent biosensor could be activated by different isolates, it may be advantageous to provide similar data for additional variates.

In the final figure of the paper (fig 5), neutralizing activity as measured with the reporter cell line for human serum post vaccination is shown. While these data illustrate the efficacy of the system, it would be necessary to compare the neutralization curves as measured with the reporter cell line to those generated using widely-utilized assays, such as PRNT or FRNT. Doing so may support the idea that this reporter cell line provides largely equivalent results.

Reviewer #2: Comments:

Quantitation of infectious virus - while the authors clearly demonstrate the use of their biosensor for measuring neutralising antibodies or the effects of drugs on viral replication, it is less clear how their system can be used for quantification of infectious virus. Given the lack of absolute quantification using luminescent readouts as well as the timing at which the assay is performed which will include multiple rounds of infection, no absolute quantification of viral stock can be achieved with this system as opposed to plaque assays. For example, when comparing variants it would not be possible to distinguish whether a different readout at 24 hours is due to different amounts of infectious units at time of infection or due to different replication dynamics.

This should be accurately stated in the manuscript and be rephrased in line 32 in the abstract ("quantitation of infectious virus") and line 145-146 ("quantitation of infectious virus").

Quantitation of replication - the authors state at several points in the manuscript that their reporter system quantitates "viral replication" (line 29, line 49, line 143-14, line 172, line 203), when "viral infection" would be more accurate. The experiment presented in Fig. 3F-G are indeed indicative of viral replication; however given the timing of events and the mechanisms of action for remdesivir and GC 376 which do not block infection but replication inside the cell, one would expect that initially the luciferase sensor should still get activated as PLPro expression precedes the viral mechanisms that are inhibited by either drug.

It could therefore be useful to provide some temporal dynamics for the luminescent sensor, e.g. how quickly after infection can an increase in luminescence be detected and for how long can that initial signal be detected (e.g. by adding neutralising antibodies after infection to prevent secondary infection).

Availability of materials - the authors should evaluate whether they can make the generated reporter clone more easily available to the community for example by depositing with BEI resources (https://www.beiresources.org/Home.aspx) or a similar repository.

**Part III – Minor Issues: Editorial and Data Presentation Modifications**

Reviewer #1: An introduction to SARS-CoV-2 proteases should be included to provide greater context.

Reviewer #2: Minor points:

What was the rationale for choosing clone B7 over G7 which had a stronger fold change upon infection?

For the purpose of comparison, it should be stated what MOI the viral dilutions correspond to in the experiments presented in Figure 3c-d and Figure 4c,e

Buchrieser et al. 2020 EMBO J should be cited as demonstrating syncytia formation and use of this phenomenon to detect viral infection using their "S-Fuse" cells

PLOS authors have the option to publish the peer review history of their article (what does this mean?). If published, this will include your full peer review and any attached files.

Reviewer #1: **Yes: **James D Brien

Reviewer #2: No
---

## [Editor Report · Decision Letter 1]

11 Jan 2022

Dear Dr Matheson,

We are pleased to inform you that your manuscript 'A protease-activatable luminescent biosensor and reporter cell line for authentic SARS-CoV-2 infection' has been provisionally accepted for publication in PLOS Pathogens.

Best regards,

Benhur Lee

Section Editor

PLOS Pathogens

Benhur Lee

Section Editor

PLOS Pathogens

Kasturi Haldar

Editor-in-Chief

PLOS Pathogens

orcid.org/0000-0001-5065-158X

Michael Malim

Editor-in-Chief

PLOS Pathogens

orcid.org/0000-0002-7699-2064
---

## [Editor Report · Acceptance letter]

7 Feb 2022

Dear Dr Matheson,

We are delighted to inform you that your manuscript, "A protease-activatable luminescent biosensor and reporter cell line for authentic SARS-CoV-2 infection," has been formally accepted for publication in PLOS Pathogens.

Best regards,

Kasturi Haldar

Editor-in-Chief

PLOS Pathogens

orcid.org/0000-0001-5065-158X

Michael Malim

Editor-in-Chief

PLOS Pathogens

orcid.org/0000-0002-7699-2064